# Changes in Peripapillary and Macular Vessel Densities and Their Relationship with Visual Field Progression after Trabeculectomy

**DOI:** 10.3390/jcm10245862

**Published:** 2021-12-14

**Authors:** Jooyoung Yoon, Kyung Rim Sung, Joong Won Shin

**Affiliations:** Department of Ophthalmology, College of Medicine, University of Ulsan, Asan Medical Center, Seoul 05505, Korea; cec1204@naver.com (J.Y.); sideral@hanmail.net (J.W.S.)

**Keywords:** primary open-angle glaucoma, optical coherence tomography angiography, vessel density, visual field, progression

## Abstract

The aim of this study was to determine the factors associated with visual field (VF) deterioration after trabeculectomy, including the peripapillary vessel density (pVD) and macular vessel density (mVD) changes assessed by optical coherence tomography angiography (OCT-A). Primary open-angle glaucoma patients with more than two years of follow-up after trabeculectomy were included. pVD was calculated in a region defined as a 750 μm-wide elliptical annulus extending from the optic disc boundary. mVD was calculated in the parafoveal (1–3 mm) and perifoveal (3–6 mm) regions. VF deterioration was defined as the rate of mean deviation (MD) worse than −1.5 dB/year. The change rates of pVD and mVD were compared between the deteriorated VF and non-deteriorated VF groups. The factors associated with the rate of MD were determined by linear regression analyses. VF deterioration was noted in 14 (21.5%) of the 65 eyes that underwent trabeculectomy. The pVD (−2.26 ± 2.67 vs. −0.02 ± 1.74%/year, *p* ≤ 0.001) reduction rate was significantly greater in the deteriorated VF group than in the non-deteriorated VF group, while that of parafoveal (*p* = 0.267) and perifoveal (*p* = 0.350) VD did not show a significant difference. The linear regression analysis showed that the postoperative MD reduction rate was significantly associated with the rate of pVD reduction (*p* = 0.016), while other clinical parameters and preoperative vascular parameters did not show any association. Eyes with greater loss of peripapillary retinal circulation after trabeculectomy tended to exhibit VF deterioration. The assessment of peripapillary vascular status can be an adjunctive strategy to predict visual function after trabeculectomy.

## 1. Introduction

Glaucoma causes progressive structural abnormalities in the optic nerve head (ONH) and the loss of visual function [1]. The mainstay of glaucoma treatment is slowing down the disease progression by preventing further ONH damage. The lowering of intraocular pressure (IOP) is the only proven way to slow down the disease progression [2,3]. Elevated IOP is a well-known risk factor for glaucoma [4,5,6,7]; however, several vascular factors also play a role in the pathophysiology of glaucoma development and progression [8]. Previous studies have evaluated the changes of the ocular hemodynamic in the ophthalmic artery, ONH, and retinal vasculature [9,10] after medical or surgical IOP reduction.

OCT angiography (OCT-A) is a non-invasive technique that can provide quantitative and reproducible vascular information of the ONH and retina [11]. OCT-A allows the precise visualization of the retinal capillary network [12] layer by layer, and a reduction of the vessel density (VD) assessed by OCT-A showed a correlation with the structural and functional parameters in glaucoma patients [13,14,15].

Increased or stable microcirculation after the surgical lowering of IOP has been reported [16,17,18,19,20]. However, research outcomes evaluating the clinical implication of postoperative microvascular changes in association with the visual prognosis are limited. Hence, we aimed to evaluate the longitudinal peripapillary and macular microcirculation changes after the surgical IOP reduction and their association with the postoperative visual field (VF) changes.

## 2. Materials and Methods

In this retrospective observational study, we recruited primary open-angle glaucoma (POAG) patients who underwent trabeculectomy by a single surgeon (KRS) at the glaucoma clinic of Asan Medical Center (Seoul, Korea) between November 2016 and January 2019. All the study procedures were performed in accordance with the principles of the Declaration of Helsinki. The Institutional Review Board of Asan Medical Center approved this study. The requirement for written informed consent was waived due to the retrospective design.

### 2.1. Participants

At the baseline examination, all the participants underwent complete ophthalmologic examinations; best-corrected visual acuity (BCVA), refractometry, slit-lamp biomicroscopy, Goldmann applanation tonometry, gonioscopy, stereoscopic optic disc/retinal nerve fiber layer (RNFL) photography, ultrasound pachymetry, standard automated perimetry (Humphrey Field analyzer with Swedish Interactive Threshold Algorithm standard 24-2 test; Carl Zeiss Meditec, Dublin, CA, USA), and OCT-A (AngioVue, Optovue Inc., Fremont, CA, USA).

The inclusion criteria of this study were as follows: patients diagnosed with POAG, with BCVA of logMAR +0.30 (Snellen 20/40) or better, a spherical refraction of –8.0 to +3.0 diopters (D), a cylinder correction within ±3 D, and clear ocular media. POAG was defined as having an open angle on gonioscopy, RNFL defects, or glaucomatous optic disc changes (neuroretinal rim thinning, disc excavation, or disc hemorrhage), and corresponding VF defects. Participants with any ophthalmic or neurological disease other than glaucoma that can affect ONH were excluded. If both eyes met the inclusion criteria, one eye was selected at random. The participants were followed-up for ≥2 years after the trabeculectomy.

Trabeculectomy was performed in patients with progressive glaucomatous changes that could not be controlled with maximum tolerated medical therapy (MTMT) and in those with an elevated IOP that could cause additional ONH damage. A single experienced glaucoma specialist (KRS) performed all surgical interventions. All glaucoma medications were continued up to the time of the surgery. Eyes with persistent hypotony maculopathy after trabeculectomy or eyes with other macular abnormalities, such as an epiretinal membrane or age-related macular degeneration, were excluded from the study. Patients who received other intraocular surgery during the 2-year follow-up period were also excluded from the study.

### 2.2. Trabeculectomy

Trabeculectomy was performed by a single surgeon (KRS). A 6- to 7-mm horizontal incision was made in the superior area, and the conjunctiva and Tenon’s capsule were carefully dissected for preparation of the fornix-based conjunctival flap. A limbus-based half-thickness scleral flap (2.5 × 2 mm) was then prepared. 0.2% mitomycin C-soaked sponge was applied under the sub-Tenon space for 2 min and copious irrigation with balanced salt solution (BSS) was performed in order to wash out the mitomycin C. The sclerectomy was made with the Kelly Descemet punch under the partial scleral flap, and the peripheral iridectomy was performed through the sclerectomy site. The scleral flap was closed with a single 9-0 nylon suture. The conjunctiva and Tenon’s capsule are secured with a 8-0 vicryl interrupted suture followed by running sutures. Bleb elevation and integrity of the conjunctival closure were checked. Topical corticosteroid (1.0% prednisolone), cycloplegics, and an antibiotic (0.5% moxifloxacin) were prescribed for approximately 1 month postoperatively, depending on the eye condition.

### 2.3. VF Assessment

Only the reliable VF test results were included in the analysis. A reliable VF test result was defined by the presence of false-positive errors < 15%, false-negative errors < 15%, and fixation loss < 20%. A glaucomatous VF defect was defined as the presence of a cluster of three or more non-edge contiguous points on a pattern deviation plot with a *p*-value < 5% (one of which had a *p*-value < 1%), confirmed by at least two consecutive examinations; pattern standard deviation with a *p*-value < 5%; or glaucoma hemifield test result outside normal limits. The VF was assessed before the surgery, and 6 months, 1 year, 1.5 years, and 2 years postoperatively. VF progression was defined as the rate of mean deviation (MD) worse than −1.5 dB annually [21]. At least five qualified VF examinations were required to be included.

### 2.4. OCT-A Imaging

The AngioVue OCT-A imaging system enables non-invasive visualization of the ophthalmic microvasculature. The dynamic motion of the moving particles, such as red blood cells, was captured using this system, with a split-spectrum amplitude-decorrelation angiography algorithm used to identify the perfused vessels. In this study, the peripapillary vasculature was measured in a 4.5 × 4.5 mm region centered on the optic disc, and within a slab from the internal limiting membrane to the posterior border of the RNFL. The peripapillary VD (pVD) was calculated in a region defined as a 750 μm-wide elliptical annulus extending from the optic disc boundary. The macular vasculature was measured in a 6.0 × 6.0 mm region centered on the fovea, and within a slab from the internal limiting membrane to the posterior border of the inner plexiform layer. The macular VD was calculated in the parafoveal and perifoveal regions, defined as concentric circular areas with an inner and outer diameter of 1 mm and 3 mm, and 3 mm and 6 mm, respectively. All the scans were evaluated for quality. The reasons for exclusion were poor image quality, defined as the signal strength index < 48; poor clarity; localized weak signals caused by artifacts, such as floaters; residual motion artifacts visible as irregular vessel patterns or disc boundaries on the en face angiogram; or segmentation failure. At least five qualified OCT-A examinations were required to be included. The circumpapillary retinal nerve fiber layer (RNFL) thickness and ganglion cell complex (GCC) thickness were also assessed using the same device.

### 2.5. Statistical Analysis

The normality was tested using the Kolmogorov-Smirnov test for all continuous variables. The baseline clinical characteristics were compared between the eyes with relatively stable VF and those with deteriorated VF after trabeculectomy using either an independent t-test or Mann-Whitney test as appropriate. A linear mixed model was performed to determine the postoperative change rate of IOP, and the OCT-A driven vascular and structural parameters. Repeated measures ANOVA was used to compare the postoperative changes of each parameter. Univariate and multivariate linear regression analyses, including the OCT-A driven vascular parameters, were used to determine the factors associated with the VF MD change rate after the trabeculectomy. A *p*-value < 0.1 in the univariate analysis was included in the multivariate analysis. The statistical analysis was performed using the SPSS software, version 20 (IBM Corp., Armonk, NY, USA). *p*-value ≤ 0.05 was considered statistically significant.

## 3. Results

Of the 81 eyes of 81 patients initially enrolled, 16 eyes were excluded due to the poor quality, poor clarity, and segmentation failure of the OCT-A image. Sixty-five eyes of 65 patients (38 male, 27 female) with successful trabeculectomy and qualified visual field tests were included in the final analysis. The baseline characteristics are described in Table 1. Postoperative VF deterioration was observed in 14 eyes (21.5%) when assessed for two years postoperatively. The preoperative clinical characteristics, including age, VF MD, and IOP did not differ between the two groups. VD parameters, RNFL, and GCC thickness, which were assessed before surgery, also did not differ between the groups. No significant postoperative complications, such as prolonged hypotony and inadequate IOP control, were observed in both groups. Topical medications were prescribed based on the patients’ postoperative condition. 1.59 ± 1.03 topical medications were used in the total population after the surgery, while 1.57 ± 1.02 in the progression group, and 1.59 ± 1.04 in the non-progression group were used, respectively. The difference in the number of postoperative topical medications between the progression and non-progression groups was not statistically significant (*p* = 0.957). 

The postoperative change rate of clinical parameters is described in Table 2 and Table 3, and Figure 1. The VF MD rate of the deteriorated VF group was −2.46 ± 0.77 dB/year, while that of the non-progression VF group was 0.06 ± 0.89 dB/year (*p* < 0.001). The rate of pVD (*p* ≤ 0.001) and RNFL thickness (*p* = 0.039) differed between the two groups, while foveal VD (*p* = 0.054) showed marginal difference. The parafoveal VD (*p* = 0.267) and perifoveal VD (*p* = 0.350) did not show significant difference. The rate of IOP (*p* = 0.672) and other structural parameters, such as GCC (*p* = 0.198) thickness, did not differ between the groups.

The baseline clinical characteristics and postoperative change rates of each OCT-A parameter were analyzed using univariate and multivariate linear regression analyses to determine the factors associated with the VF MD change rate after trabeculectomy (Table 4). None of the preoperative parameters showed an association with the postoperative VF MD rate. However, the rate of pVD (*p* = 0.006) and foveal VD (*p* = 0.057) in the univariate analysis showed a possible association with the postoperative VF MD rate; additionally, the multivariate analysis revealed that only the postoperative reduction rate of pVD showed a correlation (*p* = 0.016).

A representative case example is shown in Figure 2. A 43-year-old woman with POAG exhibited a gradual loss of peripapillary microcirculation after trabeculectomy along with progressive glaucomatous VF progression (a). A 25-year-old woman with POAG showed stable peripapillary retinal microcirculation and VF after trabeculectomy when assessed for two years with a similar level of preoperative VF MD and IOP reduction after trabeculectomy (b).

## 4. Discussion

Our study demonstrated a two-year VF change after trabeculectomy in POAG eyes and the factors associated with the postoperative VF deterioration. The deteriorated VF group showed a faster rate of pVD than the non-deteriorated VF group. Furthermore, the rate of pVD was the only factor associated with that of VF MD. The baseline clinical characteristics or the change rates of other structural parameters, or macular area VD were not relevant to postoperative VF changes.

Trabeculectomy is the most commonly performed glaucoma surgery in patients with inadequate IOP control or progression of glaucoma despite MTMT. Trabeculectomy can slow the rate of glaucomatous deterioration; however, it does not completely stop the disease progression in the long term, and some studies reported the continuous deterioration of VF despite successful IOP control after trabeculectomy [22,23,24]. Hence, factors other than IOP should be sought and considered for the care of glaucoma patients who have undergone trabeculectomy. This study demonstrated the association between the postoperative pVD change and VF MD rates, which indicates that peripapillary retinal circulation change could provide insight into postoperative VF change in glaucoma patients.

Studies [16,17,18,19,20,25] have evaluated the change of peripapillary and retinal microvasculature following a large amount of IOP reduction in glaucoma patients. Some studies [17,18,20] reported limited or no significant VD change, while others [16,19,25] have documented microvascular improvement after IOP reduction. Zeboulon et al. [20] reported a limited change in the whole peripapillary VD change and increased focal peripapillary vascular loss 1 month after deep nonpenetrating sclerotomy. Lommatzsch et al. [18] reported that no significant changes were detectable in the papillary or macular VD, RNFL, or macular ganglion cell layer thickness after trabeculectomy in open-angle glaucoma patients. Kim et al. [17] showed no significant change in the microcirculation of the peripapillary retina and choroid 3 months postoperatively after trabeculectomy in POAG patients. Contrarily, Shin et al. [19] showed that 19 (61.3%) of 31 eyes exhibited peripapillary microvascular improvement in the circumpapillary capillary dropout area three months after the trabeculectomy. Hollo et al. [16] also observed pVD improvement after large IOP reduction by topical medication in young patients with high untreated IOP. Liu et al. [25] reported the increase of vessel densities in the optic nerve head and the peripapillary area after applying prostaglandin analog for more than three weeks in the treatment-naïve eyes.

These studies demonstrated peripapillary and macular VD changes after trabeculectomy or large IOP reduction during a relatively short follow-up period (three to six months). However, in this study, we followed up with the patients who underwent trabeculectomy for two years and investigated the correlation between VD change and VF deterioration. To the best of our knowledge, this is the first study describing the association of VD with VF outcome after trabeculectomy. The diagnostic potential of macular and peripapillary OCT-A-determined VD loss in early glaucoma has been reported [11,15,26]. Along with these studies, our study showed that the assessment of peripapillary retinal circulation can be used as a predictor of visual function after IOP lowering surgery. GCC thickness change rate and the change rates of other vascular parameters did not show any difference between the deteriorated and the non-deteriorated VF groups and were not related to the postoperative VF MD change rate. Considering that most glaucoma patients who have undergone trabeculectomy already have a substantial loss of VF, a biomarker reflecting visual function change is important in the care of advanced glaucoma patients.

It is unclear why some eyes showed a decrease of peripapillary VD despite successful IOP reduction. A significant IOP decrease after trabeculectomy is known to reduce the depth of the lamina cribrosa [27,28,29]. Additionally, this level of LC depth reduction has shown an association with microvascular improvement after trabeculectomy [17,19]. However, in another study on the long-term shape and depth of LC after trabeculectomy, although most eyes showed long-term flattening and shallowing of the LC, some eyes showed a deepened LC from the baseline. Therefore, the authors concluded that a reduction of IOP plays an important role in the early phase of LC change; however, LC remodeling may play a crucial role in a stable IOP in the later phase [30]. Hence, such remodeling of LC may lead to VD reduction and glaucomatous VF deterioration.

This study has some limitations. First, we excluded 16 eyes due to the poor image quality, poor clarity, and segmentation failure of OCT-A. This could be a limitation of the current OCT-A technology. Second, the study population was of a single ethnicity; thus, the results may not be directly applied to other ethnic groups. Third, the sample size was relatively small, suggesting the need for further study with a larger population. Lastly, the follow-up period was two years postoperatively, which may be relatively short considering that glaucoma is a progressive degenerative disease. A study with a longer duration may confirm our findings.

## 5. Conclusions

This study demonstrated that a greater loss of peripapillary microvasculature after trabeculectomy is associated with visual function deterioration up to two years after surgery. This is the first study showing the relationship between the retinal microvasculature and visual function change after trabeculectomy. POAG patients with greater peripapillary capillary decrease after the IOP lowering surgery may experience VF deterioration; therefore, careful monitoring is warranted in these patients. Additionally, our results suggest the potential use of OCT-A measured peripapillary VD as a biomarker for predicting the visual function after trabeculectomy.

## Figures and Tables

**Figure 1 jcm-10-05862-f001:**
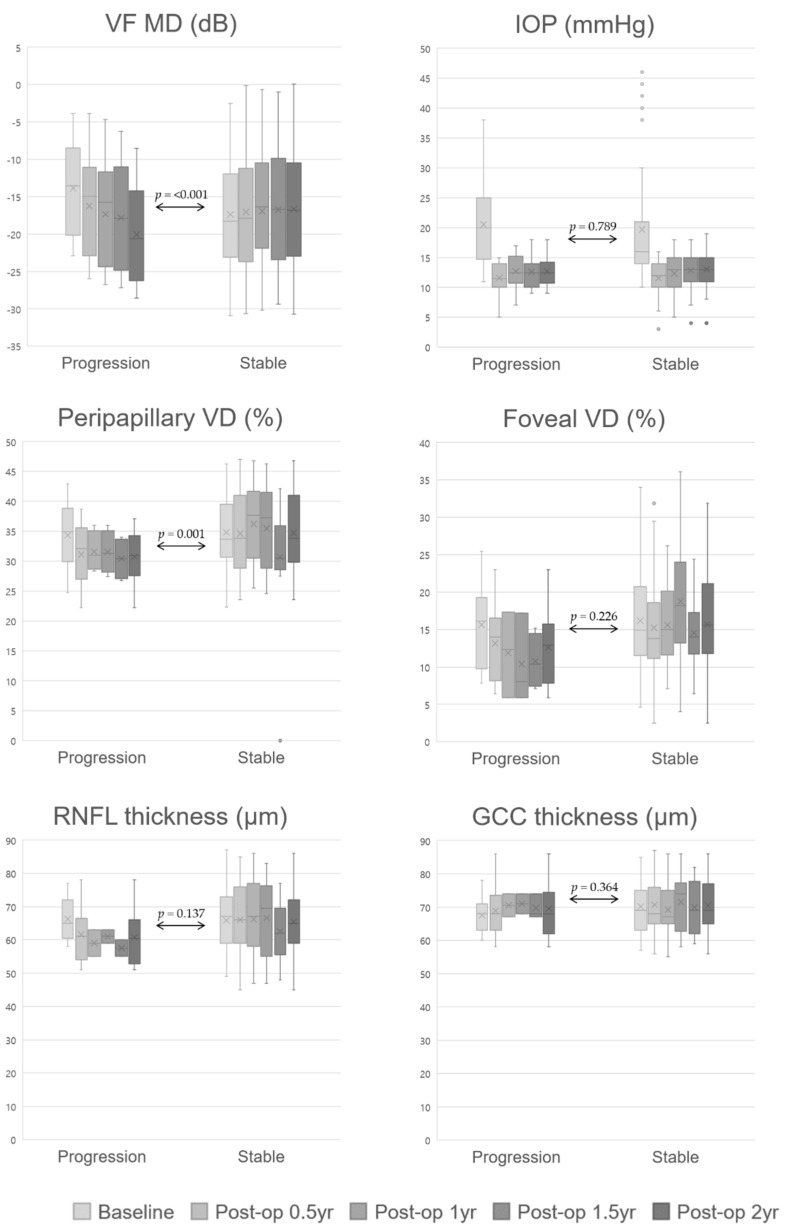
Comparison of postoperative parameter changes between the VF progression and non-progression group. Repeated measures ANOVA was used to compare the post-operative changes of each parameter, and the *p*-values of repeated measures ANOVA were presented. X: average value, horizontal line: median value, dots: outliers. Abbreviations: GCC, ganglion cell complex; IOP, intraocular pressure; RNFL, retinal nerve fiber layer; VD, vessel density; VF MD, visual field mean deviation.

**Figure 2 jcm-10-05862-f002:**
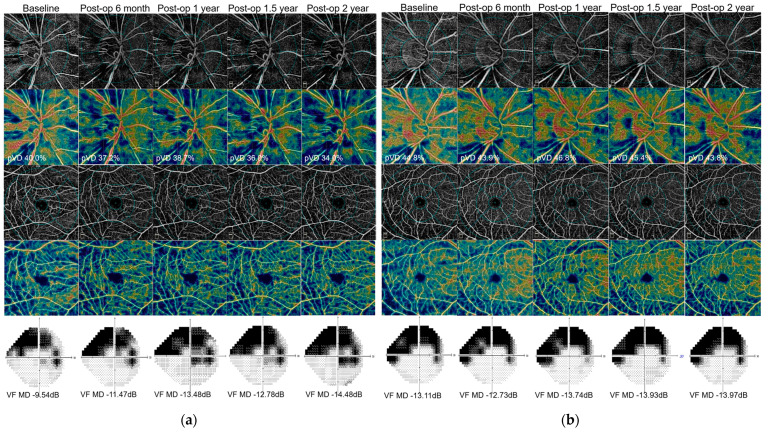
(**a**) A 43-year-old woman with POAG exhibited a gradual loss of peripapillary microcirculation after trabeculectomy along with progressive glaucomatous VF progression. (**b**) A 25-year-old woman with POAG showed stable peripapillary retinal microcirculation and VF after trabeculectomy when assessed for 2 years with a similar level of preoperative VF MD and IOP reduction after trabeculectomy. Abbreviations: POAG, primary open angle glaucoma; VF, visual field; IOP, intraocular pressure.

**Table 1 jcm-10-05862-t001:** Baseline demographics and the clinical characteristics of the VF progression and non-progression group.

Variables	Total (*n* = 65)	Progression (*n* = 14)	Non-Progression (*n* = 51)	*p*-Value *
Age (years)	54.8 ± 13.3	55.7 ± 13.5	54.6 ± 13.4	0.778
Sex, male/female	38/27	8/6	30/21	0.911
Topical medications, *n*	2.8 ± 0.6	2.6 ± 0.5	2.8 ± 0.6	0.322
Self-reported history of HTN, *n* (%)	20 (30.8%)	4 (28.6%)	16 (31.4%)	0.842
Self-reported history of DM, *n* (%)	8 (12.3%)	2 (14.3%)	6 (11.8%)	0.801
VF MD (dB)	−16.6 ± 7.9	−13.8 ± 6.7	−17.4 ± 8.1	0.137
IOP (mmHg)	19.9 ± 8.4	20.5 ± 7.9	20.0 ± 9.2	0.860
SE (D)	−2.3 ± 3.1	−3.0 ± 3.6	−2.1 ± 2.9	0.301
Axial length (mm)	24.9 ± 1.8	25.1 ± 1.8	24.9 ± 1.7	0.675
Central corneal thickness (μm)	530.1 ± 45.5	513.1 ± 51.5	533.8 ± 43.8	0.195
Peripapillary VD (%)	34.75 ± 5.85	34.25 ± 5.55	34.86 ± 5.98	0.780
Foveal VD (%)	15.99 ± 6.46	15.45 ± 5.37	16.14 ± 6.77	0.728
Parafoveal VD (%)	41.73 ± 4.99	39.77 ± 4.78	42.27 ± 4.96	0.098
Perifoveal VD (%)	37.75 ± 3.80	36.25 ± 3.53	38.16 ± 3.80	0.096
RNFL thickness (μm)	65.92 ± 8.83	66.23 ± 6.35	65.84 ± 9.41	0.889
GCC thickness (μm)	69.59 ± 7.48	67.54 ± 5.22	70.12 ± 7.92	0.271

Independent *t*-test for numerical variables; Mann-Whitney test for non-numerical variables. * *p* ≤ 0.05 was considered statistically significant. Abbreviations: HTN, Hypertension; DM, Diabetes mellitus; VF MD, Visual field mean deviation; SE, Spherical equivalence; VD, vessel density; RNFL, Retinal nerve fiber layer; GCC, Ganglion cell complex.

**Table 2 jcm-10-05862-t002:** Comparison of the postoperative change rate of the parameters between the VF progression and non-progression group.

Variables, Mean ± SD [95% CI]	Total (*n* = 65)	Progression (*n* = 14)	Non-Progression (*n* = 51)	*p*-Value *
VF MD change rate (dB/year)	−0.49 ± 1.35 (−0.82, −0.15)	−2.46 ± 0.77 (−2.91, −2.01)	0.06 ± 0.89 (−0.19, 0.30)	<0.001 *
IOP reduction rate (mmHg/year)	−3.43 ± 4.56 (−4.56, −2.30)	−3.89 ± 4.13 (−6.28, −1.51)	−3.30 ± 4.70 (−4.63, −1.98)	0.672
Peripapillary VD change rate (%/year)	−0.50 ± 2.17 (−1.04, 0.04)	−2.26 ± 2.67 (−3.81, −0.72)	−0.02 ± 1.74 (−0.51, 0.47)	<0.001 *
Foveal VD change rate (%/year)	−0.28 ± 3.55 (−1.16, 0.60)	−1.62 ± 2.52 (−3.08, −0.17)	0.09 ± 3.73 (−0.96, 1.13)	0.054
Parafoveal VD change rate (%/year)	−0.80 ± 2.97 (−1.54, −0.07)	−1.59 ± 2.71 (−3.15, −0.02)	−0.59 ± 3.03 (−1.43, 0.26)	0.267
Perifoveal VD change rate (%/year)	−0.72 ± 1.97 (−1.21, −0.23)	−1.16 ± 1.55 (−2.06, −0.27)	−0.60 ± 2.06 (−1.18, −0.02)	0.350
RNFL thickness change rate (μm/year)	−0.70 ± 3.82 (−1.66, 0.26)	−2.64 ± 3.80 (−4.93, −0.34)	−0.20 ± 3.70 (−1.25, 0.85)	0.039 *
GCC thickness change rate (μm/year)	0.59 ± 3.51 (−0.31, 1.47)	1.72 ± 3.52 (−0.41, 3.85)	0.31 ± 3.49 (−0.67, 1.29)	0.198

Independent *t*-test. * *p* ≤ 0.05 was considered statistically significant. Abbreviations: VF MD, Visual field mean deviation; IOP, intraocular pressure; VD, vessel density; RNFL, Retinal nerve fiber layer; GCC, Ganglion cell complex.

**Table 3 jcm-10-05862-t003:** Comparison of the postoperative changes of VF MD, IOP and peripapillary VD between the VF progression and non-progression group.

Variables, Mean ± SD		Total (*n* = 65)	Progression (*n* = 14)	Non-Progression (*n* = 51)	*p*-Value *
VF MD (dB)	Pre-op	−16.62 ± 7.92	−15.32 ± 5.91	−17.19 ± 8.01	<0.001 *
	Post-op 0.5yr	−16.88 ± 8.01	−17.49 ± 7.23	−16.47 ± 8.27	
	Post-op 1yr	−17.01 ± 7.81	−18.88 ± 5.88	−16.39 ± 8.06	
	Post-op 1.5yr	−16.98 ± 7.80	−19.43 ± 6.75	−16.44 ± 8.05	
	Post-op 2yr	−17.36 ± 8.16	−20.07 ± 6.55	−16.65 ± 8.10	
IOP (mmHg)	Pre-op	19.85 ± 8.42	20.50 ± 7.92	19.67 ± 8.62	0.789
	Post-op 0.5yr	11.55 ± 2.92	11.57 ± 2.53	11.55 ± 3.04	
	Post-op 1yr	12.42 ± 2.97	12.71 ± 2.89	12.33 ± 3.01	
	Post-op 1.5yr	12.78 ± 2.92	12.57 ± 2.82	12.84 ± 2.98	
	Post-op 2yr	12.98 ± 2.87	12.71 ± 2.52	13.06 ± 2.98	
Peripapillary VD (%)	Pre-op	34.75 ± 5.85	34.36 ± 5.55	34.86 ± 5.98	0.001 *
	Post-op 0.5yr	33.92 ± 6.33	31.13 ± 4.93	34.62 ± 6.51	
	Post-op 1yr	35.79 ± 6.18	30.42 ± 3.89	36.24 ± 6.50	
	Post-op 1.5yr	34.54 ± 6.62	31.18 ± 3.20	35.45 ± 6.96	
	Post-op 2yr	32.27 ± 4.22	30.28 ± 3.01	32.47 ± 4.35	

Repeated measures ANOVA * *p* ≤ 0.05 was considered statistically significant. Abbreviations: VF MD, Visual field mean deviation; IOP, intraocular pressure; VD, vessel density.

**Table 4 jcm-10-05862-t004:** Univariable and multivariable linear regression analyses to determine the factors associated with the visual field change rate after trabeculectomy.

Variables	Univariable		Multivariable (*p* < 0.1 in Univariable)	
	B ± SD	*p*-Value *	B ± SD	*p*-Value *
Age (years)	0.003 ± 0.013	0.814		
SE (D)	0.018 ± 0.055	0.739		
Central corneal thickness (μm)	0.005 ± 0.004	0.135		
Baseline IOP (mmHg)	−0.001 ± 0.019	0.961		
Baseline VF MD (dB)	−0.034 ± 0.021	0.112		
Baseline peripapillary VD (%)	−0.013 ± 0.029	0.648		
Baseline foveal VD (%)	−0.024 ± 0.026	0.369		
Baseline parafoveal VD (%)	0.009 ± 0.034	0.800		
Baseline perifoveal VD (%)	0.024 ± 0.045	0.586		
Baseline RNFL thickness (μm)	−0.013 ± 0.019	0.499		
Baseline GCC thickness (μm)	0.022 ± 0.023	0.341		
Postoperative IOP reduction rate (mmHg/year)	0.011 ± 0.037	0.771		
Postoperative peripapillary VD change rate (%/year)	0.209 ± 0.074	0.006	0.186 ± 0.075	0.016
Postoperative foveal VD change rate (%/year)	0.090 ± 0.046	0.057	0.065 ± 0.046	0.160
Postoperative parafoveal VD change rate (%/year)	0.078 ± 0.056	0.175		
Postoperative perifoveal VD change rate (%/year)	0.059 ± 0.086	0.496		
RNFL thickness change rate (μm/year)	0.075 ± 0.044	0.098		
GCC thickness change rate (μm/year)	−0.047 ± 0.048	0.335		

* *p* ≤ 0.05 was considered statistically significant. Abbreviations: SD, standard deviation; VF MD, visual field mean deviation; IOP, intraocular pressure; SE, spherical equivalence; VD, vessel density; RNFL, retinal nerve fiber layer; GCC, ganglion cell complex.

## Data Availability

The data used to support the findings of this study are included in the article, and are available on request from the corresponding author.

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
