# Peer review of "Changes in Peripapillary and Macular Vessel Densities and Their Relationship with Visual Field Progression after Trabeculectomy"

_jcm, 2021, doi:10.3390/jcm10245862_

Round 1

Reviewer 1 Report

This study assesses vessel density changes as measured by OCT-A in patients who had undergone trabeculectomy and correlates this data with the change in visual field loss. The authors have categorized their patients into two groups based on the rate of VFMD change. The rate of VFMD change has been found to correlate with the peripapillary vessel density. Although the manuscript is interesting, it requires some major revisions which are noted below.

  1. Why did the authors choose -2.0 dB/year for deteriorating visual field? This is quite a high threshold for progressive visual field loss in glaucoma. Likewise, a patient losing -1.5 dB/year (hence not fulfilling the -2.0 dB/year criterion) cannot be considered stable glaucoma. Generally, a threshold of -1.5 dB/year is preferred to define rapid progressors in glaucoma (Verma et al. Ophthalmology 2017). The authors may choose to include another analysis by choosing -1.0 dB/year or -1.5 dB/year as the cutoffs for progressive visual field loss and re-evaluate. It is also rather unclear whether a patient has to strictly fulfill this criterion at the end of the 2 years or alternatively a VF-MD loss of 1 dB at the 6-month visit qualifies for a deteriorating VF loss.
  2. The statistical method used in Table 2 is unclear. Have the authors compared the variables with respect to time using repeated measures ANOVA between the two groups? If not, I reckon this would be the most appropriate way to approach to this statistical problem.
  3. Please only mark the statistically significant differences with an asterisk in the tables.
  4. Is there any reason as to why the progressive group progressed after trabeculectomy? It would be informative to include IOPs in different timepoints as well.
  5. The OCT-A data most likely goes hand in hand with the VFMD changes. Hence it is incorrect to emphasize that OCT-A measured peripapillary vessel density is a good biomarker for “predicting” visual function. These results are affirmative for the association between visual function and blood flow parameters but insufficient to claim that the latter is predictive of the former. It is more likely that the level of postoperative IOP throughout the follow-up and the magnitude of change pre and postoperatively are the most influential determinants of both the peripapillary VD and VF-MD change after trabeculectomy. As such, it has been shown that the magnitude of IOP change significantly correlates with the change in OCT-A measured radial peripapillary capillary and optic nerve head vessel densities in treatment naïve patients who have been started on anti-glaucoma medications (Liu et al. J. Glaucoma 2017). This attests to the importance of including postoperative IOP data at the follow-up visits in the manuscript.
  6. Were the patients put on any antiglaucoma medications postoperatively?
  7. Since there are very few patients in the progressive group, ranges should also be provided for the assessed parameters.
  8. Figure 1: Please indicate what the asterisks, the “x” and the horizontal line inside the bars represent.

Author Response

Point 1: Why did the authors choose -2.0 dB/year for deteriorating visual field? This is quite a high threshold for progressive visual field loss in glaucoma. Likewise, a patient losing -1.5 dB/year (hence not fulfilling the -2.0 dB/year criterion) cannot be considered stable glaucoma. Generally, a threshold of -1.5 dB/year is preferred to define rapid progressors in glaucoma (Verma et al. Ophthalmology 2017). The authors may choose to include another analysis by choosing -1.0 dB/year or -1.5 dB/year as the cutoffs for progressive visual field loss and re-evaluate. It is also rather unclear whether a patient has to strictly fulfill this criterion at the end of the 2 years or alternatively a VF-MD loss of 1 dB at the 6-month visit qualifies for a deteriorating VF loss.

Response 1: First and foremost, thank you for the pointed remark. The authors had considered the threshold of -1.0 dB/year, -1.5 dB/year, and -2.0 dB/year, respectively, and the -2.0 dB/year criterion showed explainable results, therefore was recruited as the threshold.

The authors agree with the comment that the patients losing -1.5 dB/year cannot be considered as stable glaucoma. Additional evaluation was performed using the criteria of -1.5dB/year, and the statistical results were similar to the criteria of -2.0dB/year.

Table 1, Table 2, Figure 1 and the results were edited accordingly.

Previous (page 4, line 147): Postoperative VF deterioration was observed in 10 eyes (15.4%) when assessed for 2 years postoperatively.

Edited (page 4, line 147): Postoperative VF deterioration was observed in 14 eyes (21.5%) when assessed for 2 years postoperatively.

Previous (page 6, line 169~176): The postoperative change rate of clinical parameters is described in Table 2 and Fig-ure 1. The VF MD rate of the deteriorated VF group was -2.71±0.78 dB/year, while that of stable VF group was -0.08±0.99 dB/year (p<0.001). The rate of pVD (p=0.003) and foveal VD (p=0.046) differed between the two groups, while that of the perifoveal VD (p=0.225) and parafoveal VD (p=0.395) did not show significant difference. The rate of IOP (p=0.425) and other structural parameters, such as RNFL (p=0.059) and GCC (p=0.369) thickness, did not differ between the groups.

Edited (page 6, line 169~176): The postoperative change rate of clinical parameters is described in Table 2, Table 3, and Figure 1. The VF MD rate of the deteriorated VF group was -2.46±0.77 dB/year, while that of stable VF group was 0.06±0.89 dB/year (p<0.001). The rate of pVD (p=<0.001) and RNFL thickness (p=0.039) differed be-tween the two groups, while foveal VD (p=0.054) showed marginal difference. The perifoveal VD (p=0.267) and parafoveal VD (p=0.350) did not show signifi-cant difference. The rate of IOP (p=0.672) and other structural parameters, such as GCC (p=0.198) thickness, did not differ between the groups.

Point 2: The statistical method used in Table 2 is unclear. Have the authors compared the variables with respect to time using repeated measures ANOVA between the two groups? If not, I reckon this would be the most appropriate way to approach to this statistical problem.

Response 2: Table 2 used independent T-test to compare the change rate of each parameters. Linear mixed model was used to determine the post-operative change rate of IOP, and the OCT-A driven vascular and structural parameters. Statistical analysis of methods was edited as follows.

Previous (page 3, line 135): Linear regression analysis was performed to determine the post-operative change rate of IOP, and the OCT-A driven vascular and structural parameters.

Edited (page 3, line 135): Linear mixed model was performed to determine the post-operative change rate of IOP, and the OCT-A driven vascular and structural parameters.

Authors had performed the repeated measures ANOVA to compare the post-operative change of the variables, and concluded that the box plot figure may show the progressive post-operative change more coherently than the p-value of ANOVA. However, we are willing to add p-value of ANOVA in the figure 1, for the better explanation.

Edited (Added; page 3, line 137 & page 8, line 193): Repeated measures ANOVA was used to compare the post-operative changes of each parameters.

Edited (Added; page 8, figure 1): In Figure 1, p-value of ANOVA was inserted.

Point 3: Please only mark the statistically significant differences with an asterisk in the tables.

Response 3: The authors edited the p-values of table 1 and table 2, and only those of statistically significant differences were marked with an asterisk.

Edited (page 5, table 1 & page 6, table 2): In table 1 and table 2, only the p-values of statistically significant differences were marked with an asterisk.

Point 4: Is there any reason as to why the progressive group progressed after trabeculectomy? It would be informative to include IOPs in different timepoints as well.

Response 4: The participants did not have any post-operative IIOP events or complications that may have aggravated the glaucomatous visual field damage. This study was designed to evaluate the associated factors that explains the glaucomatous progression after the adequate surgical intervention, and as a result, vascular parameters after surgery showed association with post-op visual field change.

The authors agree that IOP change after the surgery may be informative, therefore additional table was added to show the post-operative VF-MD, IOP, and peripapillary VD change.

Edited (Added; page 6, line 188): Table 3 was added.

Point 5: The OCT-A data most likely goes hand in hand with the VFMD changes. Hence it is incorrect to emphasize that OCT-A measured peripapillary vessel density is a good biomarker for “predicting” visual function. These results are affirmative for the association between visual function and blood flow parameters but insufficient to claim that the latter is predictive of the former. It is more likely that the level of postoperative IOP throughout the follow-up and the magnitude of change pre and postoperatively are the most influential determinants of both the peripapillary VD and VF-MD change after trabeculectomy. As such, it has been shown that the magnitude of IOP change significantly correlates with the change in OCT-A measured radial peripapillary capillary and optic nerve head vessel densities in treatment naive patients who have been started on anti-glaucoma medications (Liu et al. J. Glaucoma 2017). This attests to the importance of including postoperative IOP data at the follow-up visits in the manuscript.

Response 5: Thank you for your insightful remark. As mentioned in the reviewer’s comment and the discussion of this study, the relationship of VF-MD, IOP, and peripapillary VD has been well established through previous studies. The magnitude of IOP change may be the factor for the change in peripapillary capillary vessel densities, as well as the change in visual field damage. The participants of this studies, however, did not show significant difference in the post-operative change of IOP between the stable and the progressive group, as shown in the additional table (table 3).

Although the cause and effect relationship between the VF-MD and peripapillary VD may require further investigation, the result of this study explains the importance of monitoring peripapillary VD in the post-operative patients and the potential usage as a biomarker for visual function.

Point 6: Were the patients put on any antiglaucoma medications postoperatively?

Response 6: The patients were prescribed with anti-glaucoma medications depending on the clinical condition after the trabeculectomy. 1.6 ± 1.0 topical medications were used in the total population, progressive group, and stable group, respectively. Information on post-operative anti-glaucoma medications was added on the results.

Edited (Added; page 4, line 152): Topical medications were prescribed depending on the post-operative condition; 1.6 ± 1.0 topical medications were used in the total population, progressive group, and stable group, respectively.

Point 7: Since there are very few patients in the progressive group, ranges should also be provided for the assessed parameters.

Response 7: To provide the ranges of each parameters, 95% confidence interval of each parameters of table 2 was added.

Edited (Added, page 6, table 2) [95% CI] was added.

Point 8: Figure 1: Please indicate what the asterisks, the “x” and the horizontal line inside the bars represent.

Response 8: Thank you for your remark. Explanations were added in the caption of figure 1.

Edited (Added, page 8, line 194) X: average value, horizontal line: median value, asterisks: outliers.

Reviewer 2 Report

This is an original paper of good quality and well written. 
There is an emerging evidence regarding the use of the OCT-A and glaucoma. 
This is the first study which correlates the peripapillaty vasculature changes happening after trabeculectomy with visual function.
OCT-A will be an important test to tell us about possible progression of glaucoma after surgery. 
This makes the article very interesting. 

Author Response

Point 1: This is an original paper of good quality and well written.

There is an emerging evidence regarding the use of the OCT-A and glaucoma.

This is the first study which correlates the peripapillaty vasculature changes happening after trabeculectomy with visual function.

OCT-A will be an important test to tell us about possible progression of glaucoma after surgery.

This makes the article very interesting. 

Response 1: Thank you for your kind remark. The authors are looking forward to publishing this study in the Journal of Clinical Medicine.

Reviewer 3 Report

The manuscript is well-written and interesting. In my opinion, the results section should be improved by comparing the study with other already published, not just citing them. In particular, pointing out the long follow-up and the surgical technique. In the methods section, the surgical technique should also be described.

Author Response

Point 1: The manuscript is well-written and interesting. In my opinion, the results section should be improved by comparing the study with other already published, not just citing them. In particular, pointing out the long follow-up and the surgical technique. In the methods section, the surgical technique should also be described.

Response 1: Thank you for your kind remark. Authors agree with the point that surgical technique and treatment modality must be mentioned in the article, therefore we added the trabeculectomy surgical procedure in the methods section.

Edited (Added; page 3, line 116~129): 2.4. Trabeculectomy

Trabeculectomy was performed by the single surgeon (KRS). A 6- to 7-mm horizontal incision was made in the superior area, and the conjunctiva and Tenon’s capsule were carefully dissected for preparation of the fornix-based conjunctival flap. A limbus-based half-thickness scleral flap (2.5 mm × 2 mm) was then prepared. 0.2% mitomycin C soaked sponge was applied under the sub-Tenon space for 2 minutes and copious irrigation with balanced salt solution (BSS) was performed in order to wash out the mitomycin C. The sclerectomy was made with the Kelly Descemet punch under the partial scleral flap, and the peripheral iridectomy was performed through the sclerectomy site. The scleral flap was closed with a single 9-0 nylon suture. The conjunctiva and Tenon's are secred with a 8-0 vicryl interrupted suture followed by running sutures. Bleb elevation and integrity of the conjunctival closure were checked. Topical corticosteroid (1.0% prednisolone), cycloplegics, and an antibiotic (0.5% moxifloxacin) were prescribed for approximately 1 month postoperatively, depending on the eye condition.

Round 2

Reviewer 1 Report

Please indicate what the p values in Figure 1 represent. I believe this p value is looking at the overall comparison of repeated measures between the two groups, since pairwise comparisons have not been provided.

Please also indicate what the dots in Figure 1 represent. I believe they are the outlier data.

Please find another name for "stable" disease as I have noted in my previous report, these eyes cannot be considered stable if the cutoff is 2.0dB/year for the progressive group.

"Topical medications were prescribed depending on the post-operative condition; 1.6 ± 1.0 topical medications were used in the total population, progressive group, and stable group, respectively." -> Does this mean that the mean numbers of topical antiglaucoma medications were identical between the two groups?

.....As such, it has been shown that the magnitude of IOP change significantly correlates with the change in OCT-A measured radial peripapillary capillary and optic nerve head vessel densities in treatment naive patients who have been started on anti-glaucoma medications (Liu et al. J. Glaucoma 2017). This attests to the importance of including postoperative IOP data at the follow-up visits in the manuscript. -> This comment had been omitted while revising the manuscript. I strongly recommend the authors cite this publication (and others that are relevant) and add a few discussion points with regards to the highly influential effect of IOP on the OCT-A measured vessel density parameters.

Author Response

Point 1: Please indicate what the p values in Figure 1 represent. I believe this p value is looking at the overall comparison of repeated measures between the two groups, since pairwise comparisons have not been provided.

Response 1: The p-values in the Figure 1 were derived from the repeated measures ANOVA. Explanation is added in the caption of figure 1.

Previous (page 9, line 194): Repeated measures ANOVA was used to compare the post-operative changes of each parameters.

Edited (page 9, line 194): Repeated measures ANOVA was used to compare the post-operative changes of each parameters, and the p-values of repeated measures ANOVA were presented.

Point 2: Please also indicate what the dots in Figure 1 represent. I believe they are the outlier data.

Response 2: Dots are indeed the outlier data. We added the explanation in the caption of figure 1.

Edited (Added, page 9, line 196) X: average value, horizontal line: median value, dots: outliers.

Point 3: Please find another name for "stable" disease as I have noted in my previous report, these eyes cannot be considered stable if the cutoff is 2.0dB/year for the progressive group.

Response 3: The authors agree with that -2.0dB/year criteria may classify the patients with the visual field progression of >-1.5dB/year into the stable group, therefore have changed the cutoff value for the progressive group, from the previous criteria of -2.0dB/year to the current -1.5dB/year. Thanks to your advice, we changed the criteria on the article.

Previous (page 3, line 93~94): VF progression was defined as the rate of mean deviation (MD) worse than -2.0 dB annually.

Edited (page 3, line 93~94): VF progression was defined as the rate of mean deviation (MD) worse than -1.5dB annually.

Previous (page 13, line 354): 21. Chauhan, B.C.; Malik, R.; Shuba, L.M.; Rafuse, P.E.; Nicolela, M.T.; Artes, P.H. Rates of glaucomatous visual field change in a large clinical population. Investigative ophthalmology & visual science 2014, 55, 4135-4143.

Edited (page 13, line 354): 21.    Verma, S.; Nongpiur, M.E.; Atalay, E.; Wei, X.; Husain, R.; Goh, D.; Perera, S.A.; Aung, T. Visual Field Progression in Patients with Primary Angle-Closure Glaucoma Using Pointwise Linear Regression Analysis. Ophthalmology 2017, 124, 1065-1071, doi:https://doi.org/10.1016/j.ophtha.2017.02.027.

Point 4: "Topical medications were prescribed depending on the post-operative condition; 1.6 ± 1.0 topical medications were used in the total population, progressive group, and stable group, respectively." -> Does this mean that the mean numbers of topical antiglaucoma medications were identical between the two groups?

Response 4: The statistical analysis showed that all three groups (the total population, progressive group, and stable group) used 1.6 ± 1.0 topical medications post-operatively. For better understanding, we revised the following sentence as written below.

Previous (page 4, line 152): Topical medications were prescribed depending on the post-operative condition; 1.6 ± 1.0 topical medications were used in the total population, progressive group, and stable group, respectively.

Edited (page 4, line 152): Topical medications were prescribed depending on the post-operative condition; 1.6 ± 1.0 topical medications were used in the total population, as well as the progressive group, and the stable group, respectively.

Point 5: .....As such, it has been shown that the magnitude of IOP change significantly correlates with the change in OCT-A measured radial peripapillary capillary and optic nerve head vessel densities in treatment naive patients who have been started on anti-glaucoma medications (Liu et al. J. Glaucoma 2017). This attests to the importance of including postoperative IOP data at the follow-up visits in the manuscript. -> This comment had been omitted while revising the manuscript. I strongly recommend the authors cite this publication (and others that are relevant) and add a few discussion points with regards to the highly influential effect of IOP on the OCT-A measured vessel density parameters.

Response 5: The authors have paid attention to your insightful remark, therefore added the additional table showing the post-operative change of IOP after the trabeculectomy. The correlation of IOP reduction and the OCT-A measured vessel densities have been demonstrated in various previous reports, as mentioned in the comment and the discussion part of this study (page 11, line 329 ~ page 12, line 252).

The study mentioned in the comment (Liu et al. J. Glaucoma 2021), that described the increase of vessel densities in optic nerve head and peripapillary area after applying prostaglandin analogue in the treatment-naïve eyes, is in line with the other studies, therefore we cite this publication and introduced the study result.

Edited (Added; page 12, line 252): Liu et al. [25] reported the increase of vessel densities in the optic nerve head and the peripapillary area after applying prostaglandin analogue for more than 3 weeks in the treatment-naïve eyes.
